# Breathing, (S)Training and the Pelvic Floor—A Basic Concept

**DOI:** 10.3390/healthcare10061035

**Published:** 2022-06-02

**Authors:** Helena Talasz, Christian Kremser, Heribert Johannes Talasz, Markus Kofler, Ansgar Rudisch

**Affiliations:** 1Department of Internal Medicine, Hochzirl Hospital, 6170 Zirl, Austria; 2Department of Radiology, Medical University of Innsbruck, 6020 Innsbruck, Austria; christian.kremser@i-med.ac.at (C.K.); ansgar.rudisch@i-med.ac.at (A.R.); 3Institute of Medical Biochemistry, Medical University of Innsbruck, 6020 Innsbruck, Austria; heribert.talasz@i-med.ac.at; 4Department of Neurology, Hochzirl Hospital, 6170 Zirl, Austria; markus.kofler@i-med.ac.at

**Keywords:** breathing, high-intensity physical activity, intra-abdominal pressure, pelvic floor, pelvic floor muscles, straining

## Abstract

Background: The current scientific literature is inconsistent regarding the potential beneficial or deleterious effects of high-intensity physical activities on the pelvic floor (PF) in women. So far, it has not been established with certainty whether disparate breathing mechanisms may exert short- or long-term influence on the PF function in this context, although based on the established physiological interrelationship of breathing with PF activation, this seems plausible. Objective: To propose a basic concept of the influence of different breathing patterns on the PF during strenuous physical efforts. Methodical approaches: Review of the recent literature, basic knowledge of classical western medicine regarding the principles of muscle physiology and the biomechanics of breathing, additional schematic illustrations, and magnetic resonance imaging (MRI) data corroborate the proposed concept and exemplify the consequences of strenuous efforts on the PF in relation to respective breathing phases. Conclusion: The pelvic floor muscles (PFMs) physiologically act as expiratory muscles in synergy with the anterolateral abdominal muscles, contracting during expiration and relaxing during inspiration. Obviously, a strenuous physical effort requires an expiratory motor synergy with the PFM and abdominal muscles in a co-contracted status to train the PFM and protect the PF against high intra-abdominal pressure (IAP). Holding breath in an inspiratory pattern during exertion stresses the PF because the high IAP impinges on the relaxed, hence insufficiently protected, PFMs. It seems conceivable that such disadvantageous breathing, if performed regularly and repeatedly, may ultimately cause PF dysfunction. At any rate, future research needs to take into account the respective breathing cycles during measurements and interventions addressing PFM function.

## 1. Introduction

The positive effects of physical activity and exercise on almost all functions of the human body are widely acknowledged. However, the scientific literature suggests that the female pelvic floor (PF) may be exempt from these benefits in many cases [1,2,3]. In this regard, sports and high-intensity physical activities have been subject to debate as potential risk factors for developing PF disorders, particularly in intensely physically active women [1,2,3,4,5,6]. High intra-abdominal pressure (IAP) during strenuous activities is hypothesized to stress the PF and subsequently contribute to PF dysfunction [2,3,7,8]. However, it remains unclear why, under similar training conditions, some women develop PF disorders and others do not.

In 2005, the Pelvic Floor Clinical Assessment Group of the International Continence Society (ICS) presented a standardisation of the terminology of the pelvic floor muscle (PFM) functions and dysfunctions [9], which defined a normal function of the PFMs by muscle contraction and relaxation at appropriate times. PFM contractions lead to a circular closure of the pelvic orifices and an elevation of the PF in a ventral and cranial direction. PFM relaxation results in a release of the muscular closure mechanisms allowing for the opening of the urethra, vagina, and anus, but also in a reduction in the support of the pelvic organs, allowing for their caudal displacement together with the descent of the PF. During an IAP rise, the PFMs should contract in order to maintain the support function of the PF and to close the urethra, anus, and vagina; thus, preventing incontinence and pelvic organ prolapse. On the other hand, during micturition, defecation, or childbirth, the PFMs must relax in order to reduce the support given to the urethra, anus, and vagina and to release the closure mechanisms [9].

The standardisation of the terminology represented a milestone in the definition of PFM physiology. However, two fundamental aspects of PFM function were not considered at that time. First, the PFMs do not act in isolation but in synergy with the anterolateral abdominal muscles, and both muscle groups together are inseparably linked to breathing activities [8,10,11,12,13,14,15]. Second, the superficial and deep PFM layers do not always contract synchronously and can be separately activated during voluntary efforts and during various breathing-related tasks [16,17,18]. In recent years, knowledge has advanced to take a more holistic view of the complex synergies and functions of the PFMs in connection with breathing mechanisms. Particularly during exercise and strenuous physical activities when the respiratory needs and ventilatory efforts increase and the IAP rises, it seems worthwhile to consider whether an action is performed during the inspiratory or expiratory phases of breathing and whether the associated IAP impinges on the PFMs which are in a contracted or relaxed state.

To date, these issues cannot be clarified based on the available scientific literature. Most published studies regarding PFM functions and dysfunctions—even randomized controlled intervention trials—fail to provide information on the interrelationship between breathing and abdominal muscle function with PFM activation. Such a lack of a clear definition prevents meaningful systematic meta-analyses and reviews on this topic. Therefore, the aim of this narrative article is to propose a basic concept from a theoretical viewpoint on how PFMs contract in an expiratory synergy with the abdominal muscles to protect the PF during strenuous efforts and how they relax in an inspiratory synergy to allow atraumatic evacuation processes, if needed. The paper does not report new scientific data confirming or refuting a defined research hypothesis. Rather, the concept presented here combines recent scientific evidence obtained over the last two decades with a common knowledge of classical western medicine regarding the anatomy and functions of the thoracic diaphragm, PF, thoracic and abdominal muscles, principles of muscle physiology, and biomechanics of breathing. Schematic views and dynamic magnetic resonance imaging (MRI) support the presented concept and exemplify the involvement of the PFMs in the mechanisms of breathing and the concerted synergy between the abdominal muscles and the PFMs during physical activities.

## 2. Physiological Basics

### 2.1. Reappraisal of Physiology and Function of the Muscles Surrounding the Abdominal Cavity

Similarly to the thoracic cavity, the abdominopelvic cavity is also surrounded by dynamic multilayer muscular structures. Three layers of anterolateral abdominal muscles (external oblique, internal oblique, and transverse abdominis) and rectus abdominis muscles constitute the lateral and anterior walls, respectively. The PFMs form the base and the thoracic diaphragm forms the top [10,19]. All these muscle groups together modulate the IAP [8,10,13] and react to its changes during physiological actions, such as breathing, speaking, singing, defecation and micturition, body movements, and strenuous physical activity, by adequately timed periods of muscle contraction and relaxation, thus, corresponding to muscle fibre shortening and lengthening [9,13]. Two different contractile states of muscles can be differentiated—concentric and eccentric. During a concentric contraction, a muscle contracts and shortens, whereas during an eccentric contraction, muscle fibres lengthen but are still active and develop tension against an opposing load or against gravity [20]. Thus, during an eccentric contraction, the muscles do not completely relax. Conversely, eccentric muscle actions bring muscle fibres into an optimal length and pretension for a subsequent contraction, and therefore, add to the effectiveness of repetitive muscle actions [20,21]. Although not unequivocally supported by scientific evidence, it is conceivable that a concentrically contracted, “stiff” muscle offers more resistance against an impacting force than a muscle in a lengthened, eccentrically contracted state, or even worse, a relaxed muscle without protective tension.

The muscle groups surrounding the abdominal cavity show relevant differences in muscle composition and physiology. The deep muscles, i.e., the transverse abdominis and internal oblique in the abdominal wall [22,23], the levator ani group, and the coccygeus muscle in the PF consist of a higher proportion of slow-twitch muscle fibres than the superficial muscles [16,17,24]. Slow-twitch muscle fibres are ideally suited for long-lasting muscle activities with low force, such as postural control, involuntary closure of the body orifices, and breathing [20,25]. They are innervated by small-diameter motoneurons that are controlled by central pattern generator neurons located in the pontine brainstem [26,27], which are responsible for quiet breathing, i.e., an automatic process that works without conscious intervention during sleep [18], when anesthetized, or when awake and not specifically thinking about breathing. Conscious factors can override or modify automatic functions of the respiratory control system for a limited period of time [18].

The superficially located muscles, i.e., the external oblique and rectus abdominis muscles in the abdominal wall, and the bulbospongiosus, ischiocavernosus, and transverse perineal muscles in the PF contain a high proportion of fast-twitch fibres. Their glycolytic metabolism enables them to react to immediate energy requirements and allows for fast and strong contractions during rapid movements and strenuous efforts. These muscles are predominantly innervated by fast-conducting large-diameter myelinated alpha-motoneurons and are mainly recruited during voluntary efforts [26,28]. According to recent studies, the superficial and deep abdominal muscle layers and the PFMs do not always co-contract uniformly. Indeed, concurring with their differences in fibre composition, innervation, and biomechanical function, they may be activated separately during voluntary efforts and/or during various breathing-related tasks [16,29].

### 2.2. Involvement of Abdominal and Pelvic Floor Muscles during Quiet Breathing

Breathing is a basic vital function, essential to sustain life processes. The life-long continuous mechanical act of breathing in and out is a whole-body process, accomplished by a highly coordinated rhythmic alternation of contraction and relaxation of the muscles surrounding the thoracic and abdominal cavities, in accordance with the main respiratory function of the thoracic diaphragm [30,31]. The PFMs, which act in synergy with the anterolateral abdominal muscles, are inseparably cross-linked in their function with the thoracic diaphragm. Not without reason, both these horizontal muscular structures are similarly named in traditional medical terminology as the pelvic diaphragm for the PF in conformity with the thoracic diaphragm [14,19,32]. One obvious difference between the two diaphragms is the opposite orientation of their concavities. As a consequence, a concentric contraction and shortening of the nearly radially arranged PFMs results in an upward displacement of the perineal body [9,13,33,34], whereas a contraction and shortening of the thoracic diaphragm muscles leads to a downward displacement of the central tendon [13,14,34,35]. On the contrary, during muscle relaxation and lengthening, the PF returns in a caudal direction and the thoracic diaphragm in a cranial direction (Figure 1) [13,14,33,34].

At the onset of inspiration, the muscle fibres of the thoracic diaphragm contract via innervation by the phrenic nerve and the thoracic diaphragm flattens and descends caudally [30,31]. Inspiratory muscles, which are predominantly located in the thoracic and cervical regions (parasternal, external intercostal, scalene, and sternocleidomastoid muscles), contract in synergy with the thoracic diaphragm in order to enlarge the thoracic cavity by elevating and rotating the ribs outwards [30,35]. However, they require more energy to accomplish their work when expiratory muscles, predominantly located in the abdominal region, do not relax at the same time, thereby allowing for simultaneous enlargement of the abdominal cavity [13,34,35]. Following the inspiration phase, at the beginning of expiration, the situation reverses as follows: the thoracic diaphragm and inspiratory muscles relax, the thoracic cavity narrows, and the thoracic diaphragm moves cranially. Simultaneous concentric contractions of the PFMs and the abdominal muscles lead to an upward movement of the PF in parallel to the thoracic diaphragm, as well as to a reduction in the abdominal cavity and to a further decrease in the thoracic dimensions by moving the ribs downwards [13,30,35,36,37,38].

Notably, the thoracic and abdominal cavities are both widened during inspiration and narrowed during expiration and the diaphragms show a parallel downward movement during inspiration and a parallel upward movement during expiration; yet, the two main muscle groups are in different contractile states—the thoracic diaphragm and inspiratory muscles contract, while the expiratory muscles, including the PFMs, relax, and vice versa [13] (Figure 1).

### 2.3. Switching from Quiet to Forceful Breathing in Situations with Increased Mental and Physical Strain

Quiet breathing occurs largely independent of conscious perception [31] by means of continuous phase-locked activity of the autonomously innervated deep trunk muscles with an always-open glottis to allow airflow into and out of the lungs. However, in situations with higher sympathetic activity, throughout strenuous efforts or aerobic activities, in the course of protective reflexes, such as sneezing and coughing, or during other activities that require adaptation of the breathing, such as laughing, speaking, or singing, respiration becomes stronger [31,39,40,41,42]. To ensure the body’s increased demands for oxygen, the magnitude of both inspiratory and expiratory movements of the thoracic diaphragm increase [13,30,31,35,36]. Superficial layers of thoracic and abdominal muscles become additionally recruited to support the deep muscles in their function. Activation of the superficial inspiratory muscles in the cervical and thoracic regions [30] ensures greater thoracic expansion; thus, more inflow of air into the lungs during inspiration. The superficial abdominal muscles act powerfully on the ribs to reduce the thoracic and abdominal dimensions [7,30,39] (Figure 2). The somatic nervous system control allows them to influence the amount and pressure of expelled air in different variations [16,37,39,40].

Humans can consciously activate and contract these muscles independently [31], not only in support of expiration, but also during inspiratory phases of breathing, in fact, opposing the activity of the deep trunk muscles. Physiologically, few situations in human life require such an antagonistic muscle activity. Specifically, voluntarily supported evacuation of urine and stool or labour and childbirth are initiated by a deep inhalation to relax the deep abdominal muscles and PFMs and to open the pelvic orifices. Subsequent closure of the glottis maintains the intrathoracic volume constant and keeps the thoracic diaphragm in a stable low “inspiratory” position when concomitant concentric contractions of the superficial abdominal muscles increase the IAP and directs the force caudally towards the relaxed PFMs and opened PF orifices [43].

## 3. Straining the Pelvic Floor during Training—The Good and the Bad about Exercises and Their Relationship to Breathing

In our opinion, based on such fundamental physiological mechanisms, strenuous resistance training and intensive physical activities of any kind should be performed during an expiratory motor synergy, i.e., (1) while synchronizing forceful exertions with synergistic concentric contractions of the PFMs and all layers of the abdominal muscles, and (2) with the glottis open. Synergistic contractions of the PFMs and abdominals serve to protect the PF against high IAP and additionally to support posture and stability during stance, running, fighting, and straining, whenever necessary [8,10,44,45]. The continuously open glottis allows for air exhalation and prevents excessive intrathoracic pressure increases.

To date, no clearly defined and generally accepted guidance is available on the influence of different breathing patterns and the contractile state of the respective trunk muscle groups during physical activities [46]. The guidelines for PFM rehabilitation, although giving precise instructions on how to train the PFMs, focus on concentric muscle contractions to increase muscle strength, largely ignoring the contingent interrelationship between the respective phases of breathing and the physiological co-contraction with the abdominal muscles [47]. In contrast, sports science predominantly focuses on abdominal muscle activation in different ways to increase the IAP and to augment trunk stability during exercises. Despite detailed descriptions of how to activate and use the different abdominal muscles (e.g., during manoeuvres such as abdominal bracing and abdominal hollowing [44,48] or performing abdominal hypopressive manoeuvres) [49] and discussions about which of the techniques would stabilize the lumbar spine more efficiently, no information is available on the breathing phase in which such efforts should be performed. Instructions to co-contract the PFMs and recommendations to open the glottis during exertion are vague or absent. Although this observation cannot be substantiated by scientific studies, it is common to see not only untrained people lifting heavy objects, but also recreational athletes of both genders performing high-intensity exercises and strenuous physical activities, often initiating their efforts with a deep inhalation and subsequent closing of the glottis, thus, holding their breath during exertion. Such an inspiration pattern [43,50,51,52] results in straining down on the PF as the high IAP impinges on the PFMs, which are not concentrically contracted, hence, insufficiently protected (Figure 3). Notably, such behaviour is often referred to as a Valsalva manoeuvre, reflecting the general imprecision of the definition in this field. In its original meaning and application, namely, the Valsalva manoeuvre is an exclusively expiratory pattern to expel the air through the trachea and the open glottis against a closed nose and mouth to inflate the middle ear through the Eustachian tube [43,50].

Finally, even in medical practices, expiration is often considered the result from passive, elastic recoil of the thoracic diaphragm, the lungs, and the chest walls [30,35]. Respiratory muscle training predominantly focuses on exercising the inspiratory muscles [53,54,55], while the abdominals and, in particular, the PFMs are rarely part of such training programmes. Instructions on the correct execution of a coughing manoeuvre are also vague and many people cough forcefully without consciously co-contracting their PFMs and abdominal muscles during the expiratory phase. Strong coughs can cause transient increases in the IAP, exceeding pressures of 150 cm H_2_O [32]. Obviously, the associated high IAP bulges out the exposed PFMs and abdominal wall muscles when they are not contracted; thus, it can cause incontinence or pelvic organ prolapse [32]. On the other hand, a correctly performed coughing fit definitely increases the expiratory force and, at the same time, theoretically achieves some muscle training effects because of the coordinated strong co-contraction of the PFMs and abdominal muscles.

In addition to these situations with a breath-dependent coordinated or dyssynergic interplay between the different muscle groups, it should also be noted that some people present an increased tone and persistently concentric co-contraction of both the PFMs and the abdominal muscles not only during strenuous endurance training [3,56]. There is little data available on this topic, thus, the interpretation is rather hypothetical. Muscle hypertonicity may be the result of a defined neurological disease [57], but additionally, psychological stress, emotional disorders, and chronic pelvic pain may also increase the PFMs tone, mediated by complex neural pathways [58]. Some people may even intentionally induce long-lasting PFM and abdominal muscle contractions to improve the isolated PFM strength and to shape slim bellies. Such conditions ultimately lead to an increased basic muscle tone and difficulty in relaxing the PFMs, if needed [47,57,59]. Obviously, “over-trained” and stiff PFMs and abdominal muscles hinder free abdominal breathing and allow breathing exclusively via the thoracic and rib movements, which, in the long term, may lead to muscular pain in the shoulder and neck areas. Persistently contracted muscles may lose their capacity to relax, to return to an optimal length and pretension to allow for further contractions, and to cope with acute stress situations. Consequently, healthy, slim, and well-trained women may lose urine during acute rises in their IAP because the already stiff PFMs are not sufficiently capable of performing an additional muscle contraction in order to close the PF openings firmly and to displace the urethra, which would be necessary to avoid incontinence.

On dynamic MRI sequences, there is a striking similarity between the cyclic expansive and narrowing movements of the thoracic and abdominal cavities during breathing compared to the cyclic systolic and diastolic movements of the heart. Both seem to be very efficient by means of pumping either air or blood. In this context, it is interesting to compare the alternating breathing-dependent contractile states of PFMs and abdominal muscles with the cyclic changes in the myocardium during the cardiac cycle. During the systole, efficient muscle contractions and blood ejections are only possible when the myocardial muscle cells relax sufficiently to allow for filling of the ventricle with blood during the preceding diastole [31,60].

## 4. Effects of Straining Manoeuvres on the Pelvic Floor Visualized by Dynamic Magnetic Resonance Imaging

To exemplify the consequences of common activities on the PF and their relation to the respective phases of breathing, we have previously reported dynamic MRI findings that demonstrated movement of the thoracic diaphragm, abdominal wall, and PF in real-time and indicated the corresponding muscle contractile status during the different phases of breathing, as well as while coughing, straining, and performing the Valsalva manoeuvre [13,43]. To support the proposed concept, here, we present similar real-time recordings and apply an identical MRI protocol [13,43]. After approval of the study by the local ethics committee, a 30-year-old healthy nulliparous female volunteer with normal-functioning PFMs, according to the definitions of the ICS Pelvic Floor Clinical Assessment Group [9], and well informed on the influence of breathing on the PF, performed different straining manoeuvres in a supine position in an MRI scanner (Magnetom Avanto, Siemens, Germany). The participant was instructed to breathe quietly and then to squeeze both hands with maximum force. In the first attempt (“correct”), after taking a short breath, she volitionally contracted the PFMs and abdominal muscles and exhaled through the open glottis while strongly squeezing her hands (Appendix A). According to the expiratory effort, the PF and the thoracic diaphragm move upwards in parallel and the abdominal circumference narrows (Figure 2). Subsequently (“incorrect”), she began the manoeuvre with a deep inhalation, closed the glottis, and performed the requested hand-squeezing task while straining against the closed glottis, thus, keeping the intrathoracic volume constant and holding the thoracic diaphragm in its low position (Appendix A). During this inspiratory breathing pattern, i.e., while straining against a closed glottis, the PFMs do not contract; thus, the PF is even slightly pushed downwards due to the rising IAP (Figure 3).

The dynamic MRI images were exclusively obtained to support and better visualise the presented concept and were not created as part of another study. The investigated volunteer carried out the “incorrect” straining manoeuvres intentionally.

## 5. Discussion

The views on the role of breathing in connection to the PF have changed several times since PFM exercises were first described by the gynaecologist Arnold Kegel in the late 1940s [61]. Different and sometimes opposing opinions have been held as to whether the PFMs should ideally be contracted during inhalation or exhalation, or whether they should be contracted entirely independent of breathing. During the last century, women with symptoms of PFM dysfunction and patients with low back pain were instructed to contract the PFMs during the inspiratory phases of breathing to counteract the downward movement of the thoracic diaphragm and abdominal contents and to stabilize the lumbar spine [62,63]. Recent scientific evidence, however, has demonstrated the opposite, i.e., the physiological function of the PFMs is to simultaneously co-contract with the anterolateral abdominal muscles during expiration when the thoracic diaphragm relaxes [8,11,12]. Consequently, the former strategy was abandoned without scientific discussion. Today, scientists and therapists generally take a rather vague position concerning breathing in their programmes. Even the IUGA and ICS joint reports on terminology, assessment, and treatment of PFM function and dysfunctions, which are considered the gold standard for scientists and medical professionals dealing with these topics, fail to mention the interrelationship between the PF and breathing [47,57]. This is understandable since the increasing insistence on evidence-based medicine and evidence-based physiotherapy only allows for changes in clinical theories and dogmatic practice on the basis of unambiguous and reproducible results obtained in high-quality randomised controlled clinical trials. Unfortunately, such results are not available at the present time. A high prevalence of PF disorders in exercising females and a negative impact on affected women’s quality of life are repeatedly reported; however, studies of coping strategies or investigations to identify potential pathophysiological mechanisms are scarce [4,5,6,46,64]. With this article, the authors intend to emphasize a holistic view on how breathing is intertwined with PFM functions. The presented basic concept on the impact and potential consequences of different breathing patterns during sports and strenuous physical activity on the PF attempts to underscore one possible contributor to why strong and otherwise healthy women lose urine during exertion. From a continence point of view, it makes a difference whether certain physical exercises aimed at enhancing abdominal muscle strength and/or therapeutic interventions aimed at improving PFM functions are performed during inspiration or expiration, with the glottis closed or open, and with or without the PFMs and abdominal muscles co-contracted [64].

Due to the narrative character of this article, the authors have neither intended to cite all recent and relevant publications nor to evaluate or qualify established diagnostic or therapeutic measures and recommendations regarding PFM function and dysfunction or PFM training. However, they believe that the consideration of respiratory mechanics may serve to open different perspectives and to enhance the effects of many of such interventions. Above all, however, the authors wish to emphasize that future research on both diagnostic and therapeutic measures related to PFM (dys-) function requires specific and clear terminology and definitions of the muscular contractile status and position of the PF with respect to the different phases of breathing before obtaining measurements of any kind and relating data to the outcome of the interventions. Without these prerequisites, much of the research is not comparable, not reproducible, and risks becoming obsolete.

## 6. Conclusions

Physical activity and strenuous efforts can stress the PF, and may, if performed repetitively, ultimately cause PF dysfunction when performed during inspiration or breath-holding, because high IAP impinges on non-contracted, hence, non-protected PFMs. In contrast, it seems conceivable, based on physiological synergies, that an expiratory motor pattern during body workouts and strenuous efforts is likely to protect and even train the co-contracted PFMs and lower abdominal muscles. Future studies which imply exact definitions of respiratory phases and contractile muscle states, will contribute to enhance the usefulness and comparability of the obtained data and will serve to support or refute the herein presented concept. Finally, they will ultimately enable further development of this hypothesis, which may lead to better understanding of PF disorders and their management.

## Figures and Tables

**Figure 1 healthcare-10-01035-f001:**
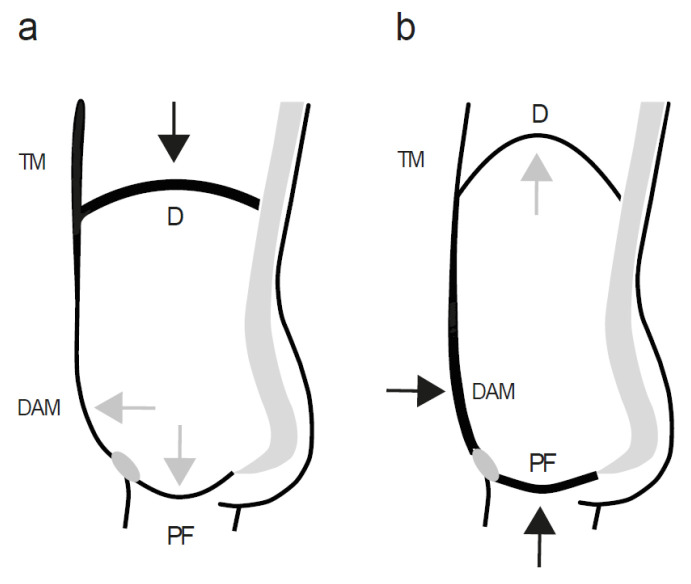
A schematic representation of a mid-sagittal view through the abdominal cavity to demonstrate the opposite orientation and contractile muscle state but parallel downward (**a**)/upward (**b**) movement of the thoracic and pelvic diaphragm where the body cavities widen and narrow during quiet breathing at rest. D = thoracic diaphragm, PF = pelvic floor, DAM = deep anterolateral abdominal muscles, TM = thoracic muscles. The thicker lines indicate a muscle contraction and the thinner lines indicate a muscle relaxation. The black arrows indicate an active force vector and the grey arrows indicate a passive force transmission.

**Figure 2 healthcare-10-01035-f002:**
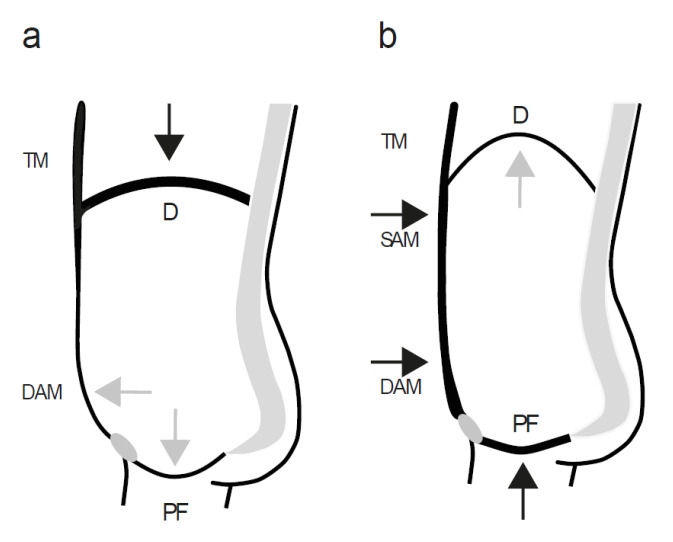
A schematic representation of a mid-sagittal view through the abdominal cavity to demonstrate the movements and contractile state of the muscle groups surrounding the abdominal cavity during (**a**) inspiration and (**b**) physical strain with a “correct” contraction of the PFMs and the anterolateral abdominal muscles during the exhaling of air through an open glottis. D = thoracic diaphragm, PF = pelvic floor, DAM = deep anterolateral abdominal muscles, SAM = superficial anterolateral abdominal muscles, TM = thoracic muscles. The thicker lines indicate a muscle contraction and the thinner lines indicate a muscle relaxation. The black arrows indicate an active force vector and the grey arrows indicate a passive force transmission.

**Figure 3 healthcare-10-01035-f003:**
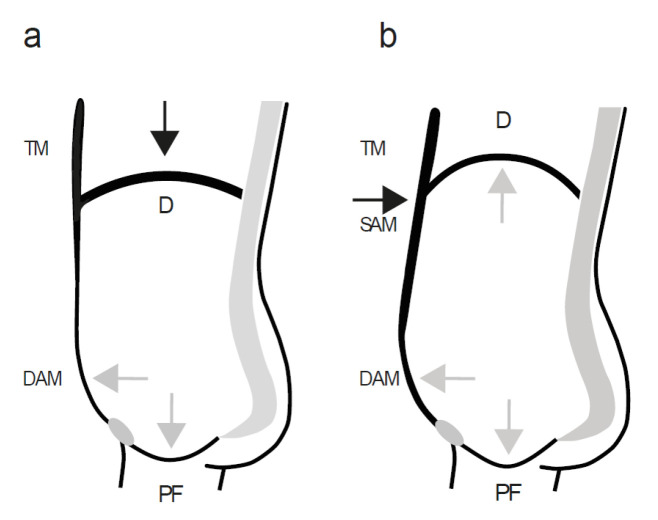
A schematic representation of a mid-sagittal view through the abdominal cavity to demonstrate the movements and contractile state of the muscle groups surrounding the abdominal cavity during (**a**) inspiration and (**b**) physical strain with an inspiratory motor synergy against a closed glottis. D = thoracic diaphragm, PF = pelvic floor, DAM = deep anterolateral abdominal muscles, SAM = superficial anterolateral abdominal muscles, TM = thoracic muscles. The thicker lines indicate muscle contraction and the thinner lines indicate muscle relaxation. The black arrows indicate an active force vector and the grey arrows indicate a passive force transmission.

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
