# Peer review of "Breathing, (S)Training and the Pelvic Floor—A Basic Concept"

_healthcare, 2022, doi:10.3390/healthcare10061035_

Round 1

Reviewer 1 Report

The authors have reviewed the effect of breathing and exercise on the pelvic floor. Although the article is well-written and well-founded, the concept they describe is very basic and does not contribute much to the development of pelvic floor research. 
They support the article with a video showing a dynamic MRI of the different phases of breathing during coughing and straining and the impact of this on the abdominal musculature and pelvic floor musculature. The effort made by the patient shown in the video cannot be extrapolated to the efforts made in the different sports fields or during daily life activities (in the video the patient is asked to clench both hands).
They should also have supported their study with other types of measurements of both the superficial and deep musculature. And they should have subjected the patient to other activities in which his muscle activation would probably be different, since to perform them he would need to use other muscle groups which, due to the appearance of synergies, could alter their statements.

Author Response

Reviewer 1

Thank you very much for the assessment of our article. Please find our answers marked in red.

The authors have reviewed the effect of breathing and exercise on the pelvic floor. Although the article is well-written and well-founded, the concept they describe is very basic and does not contribute much to the development of pelvic floor research. We fully agree with this reviewer’s objection that the concept we describe is very basic, yet we strongly believe that it has the potential to contribute to the development of further research, in that it points out the very lack of attention to breathing in much of the literature dealing with assessment and/or treatment of the pelvic floor. We are aware that there are many studies about the functions of the PF on one hand, and of the thoracic diaphragm on the other hand – both alone or in connection with the abdominal muscles. However, we are not aware of studies that clearly define the connection between all these muscle groups plus their relation with the breathing cycle. This is exactly the intention of our article: to define the well-known basics from the viewpoint of the pelvic floor (so to say), and to put them in context with different phases of breathing, in order to emphasize the necessity to define their association in future studies.

They support the article with a video showing a dynamic MRI of the different phases of breathing during coughing and straining and the impact of this on the abdominal musculature and pelvic floor musculature. The effort made by the patient shown in the video cannot be extrapolated to the efforts made in the different sports fields or during daily life activities (in the video the patient is asked to clench both hands). In our opinion, squeezing the hands strongly is indeed a simple type of strength training. However, as this manoeuvre can be performed in inspiration as well as in expiration, it can indeed be extrapolated to other tasks in sports and daily life, which can also be performed in inspiration and expiration. The completely disparate effect on the PF should principally be comparable. Of course, we are aware that different sport activities may exert different magnitudes of impacts upon the pelvic floor; however, our intention was to point out a principal mechanism, which is clearly disparate in inspiration and expiration. Furthermore, in the limited space of an MRI scanner, not many motor tasks can readily be performed without negatively affecting the recording process.

They should also have supported their study with other types of measurements of both the superficial and deep musculature. And they should have subjected the patient to other activities in which his muscle activation would probably be different, since to perform them he would need to use other muscle groups which, due to the appearance of synergies, could alter their statements. This objection is very understandable, and we hope that this reviewer will accept our response: We have in the past performed various dynamic MRI examinations predominantly in healthy people during various activities. We have previously published results on breathing, coughing, straining, and performing a Valsalva manoeuvre. We did not intend to repeat already published findings. Moreover, the most important motivation of this article was to proclaim a basic concept about how breathing and pelvic floor muscle function are related to set the stage for future research, rather than to present results of a scientific study. We are sure that the proposed concept will be scientifically discussed (after its publication), and future studies will be published to support or refute it. To us it seems most important that future studies exactly define in which respiratory phase and in which contractile muscle state measurements and results are obtained in order to enhance their comparability. We hope that the presented concept may be a basis for future discussions.

Reviewer 2 Report

This is a very important research with clinical impact. The only comment and suggestion would be to include the last ICS terminology paper. 

Author Response

This is a very important research with clinical impact. The only comment and suggestion would be to include the last ICS terminology paper.  We thank this reviewer very much for his/her positive assessment of the presented basic concept. We have included the most recent ICS terminology paper in the manuscript (Ref 47)

Reviewer 3 Report

New hypotheses and theories are often difficult to publish. One of the reasons for this is that authors may become too close to their ideas and not follow a scientific process to inform the reader how they came to this hypothesis. In this case, you have presented a narrative paper, rather than a step by step development of your theory in a scientific way.

Your theory needs to be workshopped more before it can be interesting to the scientific world. At this stage it is still simplistic and lacks detail. The MRI images are illustrative, but re of a healthy young subject. How does your hypothesis stand up to testing in an athlete who has been using motor patterns which you suggest do cause pelvic floor dysfunction?

In addition, your conclusion is stated in the manner of an intervention study rather than concluding that further development of this hypothesis could lead to better understanding of pelvic floor disorders and their management.

Rework the paper and test your theory in more circumstances, so it stands up to scientific scrutiny. However, well done for your ideas and being willing to put them out into the public.

Author Response

Thank you very much for the assessment of our article. Please find our answers marked in red.

New hypotheses and theories are often difficult to publish. One of the reasons for this is that authors may become too close to their ideas and not follow a scientific process to inform the reader how they came to this hypothesis. In this case, you have presented a narrative paper, rather than a step by step development of your theory in a scientific way.

Your theory needs to be workshopped more before it can be interesting to the scientific world. At this stage it is still simplistic and lacks detail. The MRI images are illustrative, but re of a healthy young subject. How does your hypothesis stand up to testing in an athlete who has been using motor patterns which you suggest do cause pelvic floor dysfunction?

In addition, your conclusion is stated in the manner of an intervention study rather than concluding that further development of this hypothesis could lead to better understanding of pelvic floor disorders and their management. We are grateful for the suggested sentence, that we have included in the conclusion.

Rework the paper and test your theory in more circumstances, so it stands up to scientific scrutiny. However, well done for your ideas and being willing to put them out into the public. We are grateful to this reviewer for their critical yet benevolent assessment of our article, which has helped us to improve our manuscript. We understand his/her concerns of course. However, our aim was not to perform a formal study presenting new scientific data, but rather to emphasize that it seems timely and mandatory to set the stage anew before gathering new data. We did not intend to define a new hypothesis nor a new theory. We intended to summarize a holistic concept based on proven physiological knowledge. In that, our paper is indeed narrative, and we have added this description in the introduction of the manuscript. It was our main aim to present a focussed scientific review of the topic (albeit incomplete), and stated in the introduction that most published studies regarding PFM functions and dysfunctions – even randomized controlled intervention trials – fail to provide information on the interrelationship of breathing and abdominal muscle function with PF activation. Such a lack of clear definition prevents meaningful systematic meta-analyses and reviews on that topic.

We hope that the proposed concept will be a basis for scientific discussion, and foremost we hope that future studies will define exactly in which respiratory phase and in which contractile muscle state PF measurements and results are obtained in order to enhance their usefulness and comparability, and to support or refute our concept.

We agree that the conclusion may be too deterministic. We have toned down our conclusive statement and have added a future outlook as suggested.

Reviewer 4 Report

The aim of this article was to propose a basic concept from a theoretical viewpoint on how

pelvic floor muscle contract in expiratory synergy with abdominal muscles to protect the pelvic floor during strenuous efforts, and how they relax in inspiratory synergy to allow atraumatic evacuation processes. The manuscript was a good-written review and addressed an important issue. I have a few comments as below.

  1. The review should provide the strategy and range of article search and review, so that we can have a better understand the method of this manuscript.
  2. As concluded, physical activity and strenuous efforts can stress the pelvic floor, and may cause pelvic floor dysfunction. More information about the type, intensity, duration, frequency of physical activity and exercise should be mentioned in the paper.
  3. Suggestions for intervention of respective breathing cycle during exercise should be discussed with new research and program in this area. It is very important for future study to conduct intervention to protect pelvic floor.

Author Response

Thank you very much for the assessment of our article. Please find our answers marked in red.

The aim of this article was to propose a basic concept from a theoretical viewpoint on how pelvic floor muscle contract in expiratory synergy with abdominal muscles to protect the pelvic floor during strenuous efforts, and how they relax in inspiratory synergy to allow atraumatic evacuation processes. The manuscript was a good-written review and addressed an important issue. I have a few comments as below.

  1. The review should provide the strategy and range of article search and review, so that we can have a better understand the method of this manuscript. In the introduction of the present article, we tried to explain this very problem. We did not intend to review all the pertinent literature and therefore, we have not included a description of a specific search strategy. In fact, we tried to explain why a meaningful review and/or meta-analysis of published results related to measurements and/or interventions seems impossible in the field of PFM. We tried to explain that there is a close relationship of PF function in relation to the breathing cycle with disparate contractile states and motor synergies. As long as these relationships are not taken into account, and as long as they are not described nor defined in published studies, these results cannot be compared and will not qualify for a meaningful review or meta-analysis.

  1. As concluded, physical activity and strenuous efforts can stress the pelvic floor, and may cause pelvic floor dysfunction. More information about the type, intensity, duration, frequency of physical activity and exercise should be mentioned in the paper. As now emphasized in the manuscript, we did not perform a new study, did not formulate a new hypothesis, and did not obtain new results. In contrast, we intended to describe in a narrative way what is known about the connection between breathing mechanics and pelvic floor and abdominal muscle function and what should be defined in studies dealing with this topic.

  1. Suggestions for intervention of respective breathing cycle during exercise should be discussed with new research and program in this area. It is very important for future study to conduct intervention to protect pelvic floor. We have addressed this very aspect in the revised and rephrased conclusion.

Reviewer 5 Report

Thank you for the opportunity to review this manuscript. The authors took up an interesting and challenging topic. A discussion about the associations between different forms of breathing and their impact on pelvic floor muscles is needed. I have, however, some concerns and comments I would like the authors to address.
My main concern refers to the presented point of view.
I have an impression that you are focusing mainly on harmful aspects of physical activity for PFM and pathologizing IAP. There is a big responsibility connected to publishing research and opinion papers. We must be aware that what we write may influence clinical practice. As researchers, we need to be objective, and conscious of our confirmation biases. We should restrain from imposing personal beliefs - our mission is not only to seek evidence that supports our hypotheses.
This paper (and thus readers) would benefit from a broader discussion on the IAP and PFM. I would expect that you broaden your discussion on other available perspectives. This will make your paper more complete and less biased toward one belief.
See some of the examples below:
- Lines 208-211 (that, I believe, should be referenced or stated this is your opinion):
“Based on such fundamental physiological mechanisms, strenuous resistance training and intensive physical activities of any kind should be performed during an expiratory motor synergy, i.e., (1) while synchronizing forceful exertion with synergistic concentric contraction of the PFM and all layers of abdominal muscles, and (2) with the glottis open.” Additionally, you suggest lifting through exhaling, with synchronous activation of PFM (“Instructions to co-contract PFM and recommendations to open the glottis during exertion are vague or absent.”) Hagins et al (2006) showed that max inhale followed by exhaling through a lift can result in HIGHER IAP than a breath-hold following exhalation (ie. on a small volume of air in the lungs). Pelvic floor response also matters. Some amount of PFM activation will meet the demand of increased IAP to limit bladder neck descent (Baessler et al 2017) but TOO much PFM activation can increase IAP further and result in bladder neck descent (Junginger 2018). I think you should also add more explanation on breath-hold/Valsalva vs bearing down. We generally see that closure of the glottis (as in a breath-hold) helps maintain IAP, while Valsalva (forced exhalation against that closed glottis) increases pressure further. You are writing “the high IAP impinges on PFM, which are not concentrically contracted, hence insufficiently protected”. However, there exists also another point of view. Impact activities mean that PFM not only has increased intraabdominal pressure to contend with but also the weight of internal organs bouncing up and down. One could argue that to manage this, the pelvic floor needs to be dynamically responsive - think trampoline rather than a concrete wall (meaning solid concentric contraction). Arranz-Martin et al (2021) showed that bladder base position varied with different patterns of tension (muscular activity) in the pelvic floor and abdominal wall muscles AND across different subjects. This is why I find interesting the discussion about the "rules" for what everyone should do with respect to volitional pelvic floor activation during different activities when it comes to PFM and the optimal function of the "core canister". - Similarly, you should broaden the discussion when mentioning the possible harmful consequences of physical activity.
Taking into consideration the known detrimental consequences of diminished physical activity in our patient population, we have to be cautious while concluding that physical activity may be harmful leading to PFM dysfunction. I agree that at the elite/professional level sport may have some harmful effects. But if you decide to keep this rhetoric, I would suggest you show also the other side of the story. E.g. what about physical activity that will drive adaptive changes in PFM without exceeding their current capacity? With impact activities like skipping, the pelvic floor contracts automatically and reflexively - way faster than we can intentionally control and with contractions that are stronger than what we can voluntarily perform (Leitner 2017, Moser 2018).
Apart from that, I would like to point out several minor recommendations:
- Several sections are very long and the set of references appears at the end of them e.g. lines 85-92 or 107-113. Are you sure all those references are applicable for the whole part? If not, please consider referencing the individual statements with the relevant literature.
- Lines 176-177: Again, I have some concerns about the referencing – are all those references applicable? Do all of them support the statement “To ensure the body’s increased demands for oxygen, the magnitude of both inspiratory and expiratory movements of the TD increase up to sevenfold”?
- Several statements sound speculative e.g. lines 210-211, 214-215, 227-234, 250, 268-277.
Please, add relevant references or clearly state these are your opinions/observations.
- Lines 329-331 – again, this statement should be referenced.
- Again referencing: lines 334-336 – are you sure this is a correct reference to support this statement?
- Your conclusions need revision to improve coherence with the aims. You were not assessing the influence of physical activity on PFM and the efficacy of the protective role of expiratory motor synergy.
- Is there any specific reason why you are using both abbreviations: PF (pelvic floor) and PFM (pelvic floor muscles)? It is a bit misleading for the reader. Please consider using one term/abbreviation unless you justify otherwise.
- There are unnecessary periods at the beginning of several sections (lines 32, 83, 324, 364).
- Please, consider resigning from the ‘TD’ abbreviation across the manuscript – this is not a well established abbreviation as in the case of PFM or IAP. Writing the term in full will make the text clearer and easier to follow for your readers.
- Line 325: “The role of breathing in connection to the PF has changed several times”. This sentence is not clear and needs to be modified. Do you mean here that the views/perceptions on the role of breathing in connection to the PF were changing over time? The following section (lines 327-336) sounds speculative and opinion-based. Please consider re-writing it with the use of adequate referencing or state these are your opinions.

Author Response

Thank you very much for the assessment of our article. Please find our answers marked in red.

Thank you for the opportunity to review this manuscript. The authors took up an interesting and challenging topic. A discussion about the associations between different forms of breathing and their impact on pelvic floor muscles is needed. I have, however, some concerns and comments I would like the authors to address.
My main concern refers to the presented point of view.
I have an impression that you are focusing mainly on harmful aspects of physical activity for PFM and pathologizing IAP. There are many studies, reviews, and statements from renowned experts that prove the positive effect of physical activities and PFM training on PFM function. However, to date, it remains unclear why some women under similar training conditions develop PF disorders and others do not. The aim of our article was to define a holistic concept based on the now known relationship of PFM to respiration, with different contraction states, hence fibres lengths, hence displacements of the concavely shaped diaphragms, depending on the respiratory cycle of inspiration vs. expiration, in order to explain how a muscle strain may exert “good or bad” effects by strengthening or weakening horizontal muscle groups in the body. We are aware that this may be regarded as a theoretical concept, however, it is based on known physiology. We hope to set the stage with this pioneering concept for discussions and further research.

There is a big responsibility connected to publishing research and opinion papers. We must be aware that what we write may influence clinical practice. As researchers, we need to be objective, and conscious of our confirmation biases. We should restrain from imposing personal beliefs - our mission is not only to seek evidence that supports our hypotheses.
This paper (and thus readers) would benefit from a broader discussion on the IAP and PFM. I would expect that you broaden your discussion on other available perspectives. This will make your paper more complete and less biased toward one belief.
We thank this reviewer for the wise objection. We are aware of our responsibility and we admit that our concept and attitudes may be subjectively influenced. In our clinical practice, we noticed the high prevalence of PFM dysfunction in women, and we also observed a high rate of unfavourable breathing patterns in terms of straining rather than exhaling when performing strenuous tasks, thus leading to bulging out of the abdomen and downwards displacement of the PF. Based on the physiological relationship between breathing cycle and pelvic floor activity a causal relationship seemed logical. Although we cannot prove that directly, we have indirect indications.

In attempting to write a review on the effects of PFM interventions, we became sadly aware of the lack of information on how reported interventions were related to breathing, although based on physiology this seems so important. We stated in the introduction that “ Most published studies regarding PFM functions and dysfunctions – even randomized controlled intervention trials – fail to provide information on the interrelationship of breathing and abdominal muscle function with PF activation. Such a lack of clear definition prevents meaningful systematic meta-analyses and reviews on that topic. Therefore, the aim of this article is to propose a basic concept from a theoretical viewpoint” …. We are convinced of our concept, but we do not insist that it is correct. If the concept will be accepted for further discussion, future studies will support or refute it. We will of course accept both outcomes,! To us it seems important that later studies exactly define in which respiratory phase, and in which contractile muscle state they obtain measurements and results. The concept presented here may be a basis for such a discussion.

The article is indeed intended as an opinion paper, and therefore we cannot help being subjective. We have tried to emphasize this in the discussion.

See some of the examples below:
- Lines 208-211 (that, I believe, should be referenced or stated this is your opinion):
“Based on such fundamental physiological mechanisms, strenuous resistance training and intensive physical activities of any kind should be performed during an expiratory motor synergy, i.e., (1) while synchronizing forceful exertion with synergistic concentric contraction of the PFM and all layers of abdominal muscles, and (2) with the glottis open.” This statement is the core idea of the presented concept, and it is indeed our opinion. We have changed the sentence and emphasised our subjectivity.

Additionally, you suggest lifting through exhaling, with synchronous activation of PFM (“Instructions to co-contract PFM and recommendations to open the glottis during exertion are vague or absent.”) Hagins et al (2006) showed that max inhale followed by exhaling through a lift can result in HIGHER IAP than a breath-hold following exhalation (ie. on a small volume of air in the lungs). The study by Hagins et al. is indeed of interest for the topic, because it showed that “The inhalation-hold breath pattern produced IAP that was significantly greater than that produced by both exhalation hold and inhalation–exhalation breath patterns.” Unfortunately, the authors only considered the IAP without mentioning the role of the PFM and abdominal muscles in modulating and reacting to changes in IAP. Therefore, we chose not to include the article in the reference list. In our opinion, the problem for the PF is not so much the magnitude of the IAP. Certainly, too little is still known, and large inter-individual variations are likely. In any case, however, IAP is transmitted to every part of the abdominal walls, independently of its magnitude. Obviously, muscle groups in a concentrically contracted state can resist better than muscles in an eccentrically contracted (lengthened, hence not-protected) state.

Pelvic floor response also matters. Some amount of PFM activation will meet the demand of increased IAP to limit bladder neck descent (Baessler et al 2017) but TOO much PFM activation can increase IAP further and result in bladder neck descent (Junginger 2018). We are aware of the interaction between IAP and muscle activation of the abdominal core. Obviously, PFM activation is higher when IAP rises, and otherwise IAP rises when PFM and abdominal muscles contract. We tried to address these issues in the article. However, we do not have data on IAP, and the scientific evidence seems insufficient to discuss the level of the IAP. In our opinion, it is important to define the muscular state of the different muscle groups. Further investigations are needed to measure the IAP under identical conditions. The study protocol of Junginger et al. 2018 did not consider the respective breathing phase during the PFM contractions. According to our experience, women tend to leave the glottis open and exhale during prolonged submaximal PFM contractions, whereas they inhale, close the glottis and then contract the PFM when requested to contract maximally quickly and strongly. We regret the repetition, but without a precise definition of respiratory phase and muscular contraction state, it is impossible to interpret the results of the study in full.

I think you should also add more explanation on breath-hold/Valsalva vs bearing down. We generally see that closure of the glottis (as in a breath-hold) helps maintain IAP, while Valsalva (forced exhalation against that closed glottis) increases pressure further. You are writing “the high IAP impinges on PFM, which are not concentrically contracted, hence insufficiently protected”. However, there exists also another point of view. Impact activities mean that PFM not only has increased intraabdominal pressure to contend with but also the weight of internal organs bouncing up and down. One could argue that to manage this, the pelvic floor needs to be dynamically responsive - think trampoline rather than a concrete wall (meaning solid concentric contraction). The distinction between breath-hold and performing a Valsalva manoeuvre is a delicate and difficult matter. We do not agree with the prevailing definitions and published two articles devoted to this issue (doi: 10.1007/s00192-011-1397-0 and doi: 10.1016/j.ejogrb.2012.06.019). The most critical problem is the often-used definition of the Valsalva manoeuvre as a “forced exhalation against a closed glottis”. This is historically incorrect and non-physiological! There are two possible strategies: one can either 1) exhale through an open glottis in order to inflate the middle ear (=Valsalva manoeuvre =expiratory motor synergy = the muscle groups caudal of the thoracic diaphragm are concentrically contracted) or 2) close the glottis following inspiration (=breath holding). A straining manoeuvre or any type of high-intensity physical activity performed during breath holding results in bearing down, because the muscle groups caudal of the thoracic diaphragm are eccentrically contracted (=inspiratory motor synergy).

Arranz-Martin et al (2021) showed that bladder base position varied with different patterns of tension (muscular activity) in the pelvic floor and abdominal wall muscles AND across different subjects. This is why I find interesting the discussion about the "rules" for what everyone should do with respect to volitional pelvic floor activation during different activities when it comes to PFM and the optimal function of the "core canister". - Similarly, you should broaden the discussion when mentioning the possible harmful consequences of physical activity. We are aware that we are dealing with a difficult and controversial topic in our article because there are many studies that seem to contradict our concept. But these studies (Arranz-Martin 2021 included) have not taken breathing into account. It is undisputed that there are many different functioning pelvic floors; there are also interindividual different breathing patterns without any symptoms and complaints for the individual person. It is not the aim of the article to evaluate this phenomenon. However, we would like to present a clear concept for future discussion in order to set the stage for meaningful and comparable studies in the future. We simply believe that, as there is a relationship between PF, abdominal muscles, and breathing which needs to be clearly defined in interventions addressing the PF.

Taking into consideration the known detrimental consequences of diminished physical activity in our patient population, we have to be cautious while concluding that physical activity may be harmful leading to PFM dysfunction. I agree that at the elite/professional level sport may have some harmful effects. But if you decide to keep this rhetoric, I would suggest you show also the other side of the story. E.g. what about physical activity that will drive adaptive changes in PFM without exceeding their current capacity? With impact activities like skipping, the pelvic floor contracts automatically and reflexively - way faster than we can intentionally control and with contractions that are stronger than what we can voluntarily perform (Leitner 2017, Moser 2018). We are far from concluding that physical activity may generally be harmful for the PF. A co-ordinated PFM and abdominal muscle concentric contraction during exertion whatsoever can strengthen the PFM. We tried to define and conclude the different influences in the conclusion part: Physical activity and strenuous efforts can stress the PF, and may, if performed repetitively, ultimately cause PF dysfunction, when performed during inspiration, because high IAP impinges on non-contracted, hence non-protected PFM. In contrast, an expiratory motor synergy during body workouts and strenuous efforts is likely to protect and even train the co-contracted PFM and lower abdominal muscles.” This concurs with the study results of Leitner, Moser and co-authors. They found that during running PFM were activated continuously. This suggests (the authors have not addressed breathing mechanics) that the women strongly exhaled during running. We have addressed this technique in the main text (section 3).

Apart from that, I would like to point out several minor recommendations:
- Several sections are very long and the set of references appears at the end of them e.g. lines 85-92 or 107-113. Are you sure all those references are applicable for the whole part? If not, please consider referencing the individual statements with the relevant literature. - Lines 176-177: Again, I have some concerns about the referencing – are all those references applicable? Do all of them support the statement “To ensure the body’s increased demands for oxygen, the magnitude of both inspiratory and expiratory movements of the TD increase up to sevenfold”? - Several statements sound speculative e.g. lines 210-211, 214-215, 227-234, 250, 268-277.
Please, add relevant references or clearly state these are your opinions/observations.
- Lines 329-331 – again, this statement should be referenced. We have adapted the contested statements and rearranged the in-text citations as suggested.

  • Your conclusions need revision to improve coherence with the aims. You were not assessing the influence of physical activity on PFM and the efficacy of the protective role of expiratory motor synergy. We have rephrased the conclusion.

  • - Is there any specific reason why you are using both abbreviations: PF (pelvic floor) and PFM (pelvic floor muscles)? It is a bit misleading for the reader. Please consider using one term/abbreviation unless you justify otherwise. We attempted to adhere to the terminology suggested by Messelink et al. (2005). They make a clear distinction between PF and PFM. In a few cases, we have changed PF to PFM.

- There are unnecessary periods at the beginning of several sections (lines 32, 83, 324, 364) These unnecessary periods – obviously appearing during the formatting process when submitting the manuscript – have been removed.

- Please, consider resigning from the ‘TD’ abbreviation across the manuscript – this is not a well established abbreviation as in the case of PFM or IAP. Writing the term in full will make the text clearer and easier to follow for your readers. We have replaced the abbreviation TD with thoracic diaphragm.

- Line 325: “The role of breathing in connection to the PF has changed several times”. This sentence is not clear and needs to be modified. Do you mean here that the views/perceptions on the role of breathing in connection to the PF were changing over time? The following section (lines 327-336) sounds speculative and opinion-based. Please consider re-writing it with the use of adequate referencing or state these are your opinions. Toward the end of the last century, based on misinterpretations of human physiology, at least two generations of women with symptoms of PFM dysfunction were instructed to contract PFMs during inspiration and breath holding and to relax them during expiration. This strategy failed and was abandoned without any scientific discussion. Nowadays, in clinical practice most specialists instruct patients to exhale during PFM contraction or at least to continue to breathe without breath holding.

We have modified the respective statement in the manuscript.

Round 2

Reviewer 3 Report

This manuscript has been thoroughly reviewed, adn the authors have willingly explained their position. While I don't think it is a good paper, it now merits publication and wider scientific discussion.

Reviewer 4 Report

I have no further comments.